# ADAPTIVE MULTI-HEAD CONTRASTIVE LEARNING

## ABSTRACT

In contrastive learning, two views of an original image generated by different augmentations are considered as a positive pair whose similarity is required to be high. Moreover, two views of two different images are considered as a negative pair, and their similarity is encouraged to be low. Normally, a single similarity measure given by a single projection head is used to evaluate positive and negative sample pairs, respectively. However, due to the various augmentation strategies and varying intra-sample similarity, augmented views from the same image are often not similar. Moreover, due to inter-sample similarity, augmented views of two different images may be more similar than augmented views from the same image. As such, enforcing a high similarity for positive pairs and a low similarity for negative pairs may not always be achievable, and in the case of some pairs, forcing so may be detrimental to the performance. To address this issue, we propose to use multiple projection heads, each producing a separate set of features. Our loss function for pre-training emerges from a solution to the maximum likelihood estimation over head-wise posterior distributions of positive samples given observations. The loss contains the similarity measure over positive and negative pairs, each re-weighted by an individual adaptive temperature that is regularized to prevent ill solutions. Our adaptive multi-head contrastive learning (AMCL) can be applied to and experimentally improves several popular contrastive learning methods such as SimCLR, MoCo and Barlow Twins. Such improvement is consistent under various backbones and linear probing epochs and is more significant when multiple augmentation methods are used.

## 1 INTRODUCTION

Contrastive learning is an important line of work in self-supervised learning (SSL) which offers a promising path to leveraging large quantities of unlabeled data. Its main idea is to encourage two views of the same image (positive pair) to have similar embeddings and thus a high similarity, and those of different images (negative pair) to have a low similarity. As such, the similarity measure is an important component influencing representation learning.

In literature, multiple augmentations are usually used to create a view of an image. For example, rotation, scaling, translation, and flipping are used in SimCLR (Chen et al., 2020) and MoCo (He et al., 2020). However, the use of multiple augmentations make positive pairs often look dissimilar and negative pairs occasionally similar: examples are presented in Fig. 1(a). Therefore, there exists non-negligible diversity in the distribution of similarity of image pairs. As shown in Fig. 1(b)-(d), when the number of augmentations increases from 1, 3 to 5, the similarity distributions of positive and negative pairs of the SimCLR method become more complex, *e.g.*, the similarity of more positive pairs drops below 0.5; thus, we observe increasingly significant overlapping regions, indicating compromised similarity learning.

We identify two limitations of existing methods which prevent them from addressing the above-mentioned problem. *First*, existing methods usually use a single feature projection head and a single similarity measure (Chen et al., 2020; He et al., 2020; Chen & He, 2021). While this head is supervised by standard metric loss such as contrastive loss, a single projection has a single mode of image characterization which would be insufficient to describe the diverse image content caused by multiple augmentations. A consequence is that positive pairs sometimes have low similarity scores.

*Second*, existing methods usually use a global temperature to scale similarity, which, after careful tuning, is shown to improve feature alignment and uniformity (Wang & Isola, 2020). Under this scheme, the same scaling is applied to all the pairs, which does not alleviate overlapping exhibited in Fig. 1(d).

This paper aims to address the diversity issue caused by multiple augmentations while considering the limitations in existing practice. We propose adaptive multi-head contrastive learning (AMCL): it better captures the diverse image content and gives similarity scores that better separate positive and negative pairs. In a nutshell, instead of having a single MLP and cosine similarity, AMCL uses multiple repetitive MLPs and cosine similarity measures before loss computation. Within AMCL, we design an adaptive temperature which depends on both the projection head and the similarity of the current pair. We show that the idea of multiple projection heads and adaptive temperature can be applied to popular contrastive learning frameworks and yields consistent improvements. Consistent with our motivation, we report more significant improvements when using 4-5 augmentation types than 1-2 augmentation types.

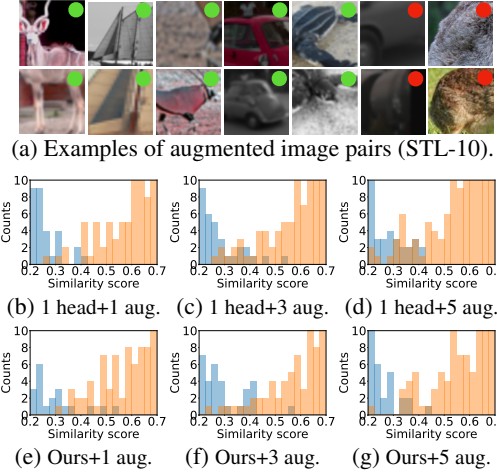

(a) Examples of augmented image pairs (STL-10).

(b) 1 head+1 aug.  (c) 1 head+3 aug.  (d) 1 head+5 aug.

(e) Ours+1 aug.  (f) Ours+3 aug.  (g) Ours+5 aug.

Figure 1: In (a), more augmentations make positive samples (green dots) look dissimilar and sometimes negative samples (red dots) similar. (b)-(d): for traditional contrastive learning methods, when increasing the number of augmentations from 1 to 5, similarities of more positive pairs drop below 0.5, causing more significant overlapping regions between histograms of positive (orange) and negative (blue) sample pairs. In comparison, our multi-head approach (e)-(d) yields better separation of positive and negative sample pairs as more augmentation types are used, *e.g.*, (g) *vs.* (d).

On the theoretical side, we derive the training objective function of AMCL based on maximum likelihood estimation (MLE). We show this objective function can be reduced to many existing contrastive learning methods, that its regularization term has interesting physical meaning, and that with it we are now able to connect temperature to uncertainty. We summarize the main points below.

i. We propose adaptive multi-head contrastive learning (AMCL) which tackles intra- and inter-sample similarity and an adaptive temperature mechanism re-weighting each similarity pair.

ii. We derive the objective function for AMCL as solution to the maximum likelihood estimation. We also discuss its mathematical insights including connecting temperature to uncertainty.

iii. Our system consistently improves the performance of a few popular constrastive learning frameworks, backbones, loss functions, and combinations of augmentation types, and is shown to be particularly useful under more augmentation types.

## 2 RELATED WORK

**Self-supervised learning** (SSL) has been as a driving force behind unsupervised learning in computer vision and natural language processing (NLP) (Balestriero et al., 2023). Its methods can be grouped into 4 broad families: deep metric learning family (Chen et al., 2020; Dwibedi et al., 2021; Du et al., 2021), self-distillation family (Grill et al., 2020; Chen & He, 2021; Caron et al., 2021; He et al., 2020; Zhou et al., 2021; Koohpayegani et al., 2021; Oquab et al., 2023), canonical correlation analysis family (Hotelling, 1936; Zbontar et al., 2021; Caron et al., 2020; Bardes et al., 2022), and masked image modeling (MIM) (Bao et al., 2022; He et al., 2022; Xie et al., 2022). Since contrastive learning and MIM can complement each other (Park et al., 2023), recent works have adopted a fusion of both to improve representation quality and transfer performance over its traditional MIM approaches (Huang et al., 2022b; Mishra et al., 2022; Wei et al., 2022; Jiang et al., 2023).

**Metric learning** is related to self-supervised learning (Balestriero et al., 2023). Commonly used similarity measurements include the triplet loss (Hoffer & Ailon, 2015), cross-entropy loss (Zhang &

Table 1: Standard contrastive learning methods and their loss functions.

| Method | Loss name | Loss function | |
|--------|-----------|---------------|---|
| SimCLR, MoCo | NT-Xent | $\ell_{\text{NT-Xent}} = -\log \frac{\exp(\text{sim}(\boldsymbol{z}_i, \boldsymbol{z}_i^+)/\tau)}{\sum_{n=1}^{N} \exp(\text{sim}(\boldsymbol{z}_i, \boldsymbol{z}_{i_n}^-)/\tau)}$ | (1.1) |
| SimSiam | Negative cos. | $\ell_{\text{SymNegCos}} = -\frac{1}{2}\text{sim}\big(\boldsymbol{z}_i, [\boldsymbol{h}_i^+]_{\text{sg}}\big) - \frac{1}{2}\text{sim}\big(\boldsymbol{z}_i^+, [\boldsymbol{h}_i]_{\text{sg}}\big)$ | (1.2) |
| Barlow Twins | Cross-corr. | $\ell_{\text{Cross-Corr}} = \sum_{l=1}^{d'}(1 - \mathcal{C}_{ll})^2 + \lambda \sum_{l=1}^{d'} \sum_{m \neq l}^{d'} \mathcal{C}_{lm}^2$ | (1.3) |
| LGP, CAN | InfoNCE | $\ell_{\text{InfoNCE}} = -\log \frac{\exp(\text{sim}(\boldsymbol{z}_i, \boldsymbol{z}_i^+)/\tau)}{\exp(\text{sim}(\boldsymbol{z}_i, \boldsymbol{z}_i^+)/\tau) + \sum_{n=1}^{N} \exp(\text{sim}(\boldsymbol{z}_i, \boldsymbol{z}_{i_n}^-)/\tau)}$ | (1.4) |

Sabuncu, 2018), and contrastive loss (Khosla et al., 2020). A contemporary work is multi-similarity learning (Mu et al., 2023), where different attribute labels of an image are used in each level of learning. Different from (Mu et al., 2023), our method is self-supervised, discusses the adaptive temperature as a useful add-on, and derives interesting mathematical insights.

**Uncertainty learning** has been studied extensively (Abdar et al., 2021; Gawlikowski et al., 2023). For example, Tao (2019) use multiple network copies trained with different parameter initializations to find various local minima. Bayesian neural networks (Goan & Fookes, 2020) and Monte Carlo dropout (Gal & Ghahramani, 2016) handle uncertainty by design, where dropout layers are equivalent to sampling weights from a posterior distribution over model parameters.

A more principled way is to capture aleatoric uncertainty (Matthies, 2007; Kendall & Gal, 2017; Hüllermeier & Waegeman, 2021) of Euclidean distance or cosine similarity, *e.g.*, heteroscedastic aleatoric uncertainty (observation noise may vary with each pair of samples). To this end, we model the maximum likelihood estimation over head-wise posterior distributions of positive samples given observations. This is a form of m-estimator (Huber et al., 1981) whose log-likelihood employs Normal distributions *a.k.a.* Welsch functions by the uncertainty estimation community.

## 3 SELF-SUPERVISED LEARNING FRAMEWORKS: A REVISIT

**Notations**. A common contrastive learning framework typically consists of a data augmentation module, a base encoder $f(\cdot)$, a projection head $g(\cdot)$ and a loss function. Stochastic data augmentation transforms a given sample randomly, resulting in two views of the same sample denoted $\boldsymbol{x}_i$ and $\boldsymbol{x}_i^+$, which are considered as a positive pair consisting of an anchor and positive sample, respectively. Their visual representations are denoted as $\boldsymbol{h}_i = f(\boldsymbol{x}_i) \in \mathbb{R}^d$ and $\boldsymbol{h}_i^+ = f(\boldsymbol{x}_i^+) \in \mathbb{R}^d$, where $d$ is feature dimension. The projection head $g(\cdot)$ maps these $d$-dim vectors to $d'$-dim vectors $\boldsymbol{z}_i = g(\boldsymbol{h}_i) \in \mathbb{R}^{d'}$ and $\boldsymbol{z}_i^+ = g(\boldsymbol{h}_i^+) \in \mathbb{R}^{d'}$, to which the contrastive learning loss is applied. Normally the multi-layer perceptrons (MLPs) are used for projection. By analogy, negative samples for anchor $\boldsymbol{x}_i$ are denoted by $\boldsymbol{x}_{i_n}^-$ ($n = 1, \cdots, N$, and $N$ is the total number of negative samples per anchor), and their features and projection head outputs are $\boldsymbol{h}_{i_n}^- = f(\boldsymbol{x}_{i_n}^-)$ and $\boldsymbol{z}_{i_n}^- = g(\boldsymbol{h}_{i_n}^-)$, respectively.

**Loss functions**. The contrastive loss function typically tries to align the anchors with their positive samples, and enlarge the distance between the anchors and their negative samples. Loss functions of some popular SSL methods are summarized in able 1. Methods such as SimCLR and MoCo use the NT-Xent loss, given in Eq. (1.1). NT-Xent is very similar to the InfoNCE loss but differs by the normalizaton step. Function $\text{sim}(\cdot, \cdot)$ in equations of Table 1 represents the cosine similarity. SimSiam uses the negative cosine similarity loss in Eq. (1.2), where $[\cdot]_{\text{sg}}$ is the stop-gradient operation. Barlow Twins in Eq. (1.3) takes a different approach by utilizing the cross-correlation loss to decorrelate the channels of both views. In Eq. (1.3), $\lambda \geq 0$ is a hyperparameter that controls the strength of decorrelation. $\mathcal{C}$ is the cross-correlation matrix computed between the outputs of two identical networks along the batch dimension, *i.e.*, $\mathcal{C}_{lm} = \sum_{n=1}^{N} z_{ln} z_{mn}^+$ (Zbontar et al., 2021).

Differing from contrastive learning, masked image modeling (MIM) learns to reconstruct a corrupted images where some parts of the image or feature map are masked out. As demonstrated in (Park et al., 2023), contrastive learning and MIM are complementary strategies. Thus, recent works,

Table 2: Loss functions of our multi-head variants of popular contrastive learning methods.

| Method | Loss function | | Regularization |
|---|---|---|---|
| SimCLR, MoCo | $\ell^{\dagger}_{\text{NT-Xent}}$ | $= \sum_{c=1}^{C} \Big( -\frac{1}{\tau_i^{c+}}\text{sim}(\boldsymbol{z}_i^c, \boldsymbol{z}_i^{c+}) + \frac{1}{\tau_{in*}^{c-}}\max_{n=1,\cdots,N}\text{sim}(\boldsymbol{z}_i^c, \boldsymbol{z}_{in}^{c-})$ | $+\beta\Omega(\tau_i^{c+}) - \beta\Omega(\tau_{in*}^{c-}) \Big)$ (2.1) |
| SimSiam | $\ell^{\dagger}_{\text{SymNegCos}}$ | $= \sum_{c=1}^{C} \Big( -\frac{1}{2\tau_i^{c+}}\text{sim}\big(\boldsymbol{z}_i^c, [\boldsymbol{h}_i^+]_{\text{sg}}\big) - \frac{1}{2\tau_i^{\widetilde{c+}}}\text{sim}\big(\boldsymbol{z}_i^{c+}, [\boldsymbol{h}_i]_{\text{sg}}\big)$ | $+\beta\Omega(\tau_i^{c+}) + \beta\Omega(\tau_i^{\widetilde{c+}}) \Big)$ (2.2) |
| Barlow Twins | $\ell^{\dagger}_{\text{Cross-Corr}}$ | $= \sum_{c=1}^{C} \Big( \sum_{l=1}^{d'}(1 - \frac{1}{\tau_l^{c+}}\mathcal{C}_{ll})^2 + \lambda \sum_{l=1}^{d'}\sum_{m\neq l}^{d'}\frac{1}{\tau_{lm}^{c-}}\mathcal{C}_{lm}^2$ | $+\beta\sum_{l=1}^{d'}\Omega(\tau_l^{c+}) - \beta\sum_{l=1}^{d'}\sum_{m\neq l}^{d'}\Omega(\tau_{lm}^{c-}) \Big)$ (2.3) |
| LGP, CAN | $\ell^{\dagger}_{\text{InfoNCE}}$ | $= \sum_{c=1}^{C} \Big( -\frac{1}{\tau_i^{c+}}\text{sim}(\boldsymbol{z}_i^c, \boldsymbol{z}_i^{c+}) + \frac{1}{\tau_{in*}^{c-}}\max_{n=1,\cdots,N+1}\text{sim}(\boldsymbol{z}_i^c, \boldsymbol{z}_{in}^{c\pm})$ | $+\beta\Omega(\tau_i^{c+}) - \beta\Omega(\tau_{in*}^{c\pm}) \Big)$ (2.4) |

LGP (Jiang et al., 2023) and CAN (Mishra et al., 2022), combine the MIM loss and the InfoNCE loss in Eq. (1.4). Kindly notice our innovations apply to the contrastive losses rather than MIM.

# 4 APPROACH

## 4.1 ADAPTIVE MULTI-HEAD CONTRASTIVE LEARNING

Typical SSL methods incorporate a projection head $g(\cdot)$, often consisting of a 2- or 3-layer MLP with ReLU activation. This projection head has proven to be highly beneficial as removing final layers of a pre-trained deep neural network helps mitigate overfitting to the training task and helps learning downstream tasks better (Balestriero et al., 2023). In AMCL we propose to **apply $C$ such projection heads**, denoted $g^1(\cdot), \cdots, g^C(\cdot)$. The goal is to capture complementary aspects of similarity between views. Our loss function takes the following general form:

$$\ell^{\dagger} = \sum_{c=1}^{C} \Big( \ell_{\text{Contrast}}\big(\boldsymbol{z}_i^c, \boldsymbol{z}_i^+, \{\boldsymbol{z}_{in}^c\}_{n=1}^N\big) + \beta\Omega(\tau_i^{c+}) - \beta\Omega\big(\{\tau_{in}^{c-}\}_{n=1}^N\big) \Big), \qquad (3)$$

$$\text{where} \quad \tau_i^{c+} = \sigma\big(\langle\phi(\boldsymbol{z}_i^c), \phi(\boldsymbol{z}_i^+)\rangle\big) \quad \text{and} \quad \tau_{in}^{c-} = \sigma\big(\langle\phi(\boldsymbol{z}_i^c), \phi(\boldsymbol{z}_{in}^-)\rangle\big).$$

In the above equation, $\beta \geq 0$ controls the temperature regularization imposed by $\Omega(\cdot)$. $\tau_i^{c+}$ and $\tau_{in}^{c-}$ denote the **learnable adaptive positive and negative temperatures**, respectively. $\langle\cdot,\cdot\rangle$ represents the dot product. $\phi(\cdot) : \mathbb{R}^{d'} \to \mathbb{R}^{d'}$ is an MLP layer[1] shared among all heads. Sigmoid function $\sigma(r) = \frac{\iota}{1+\exp(r)} + \eta$ controls the lower and upper limits of the temperature, where $\iota$ and $\eta$ are hyperparameters.

Table 2 presents our specific multihead contrastive implementations. The '$\dagger$' in Eq. (2.1)–(2.4) indicates that these losses represent our multi-head loss versions. The regularization is written as:

Figure 2: Workflow of AMCL. Given two feature vectors (extracted from the backbone) of an image pair, we first obtain their projected feature vectors through $C$ projection heads. Here, the $c$-th projection head gives us two feature vectors (anchor $\boldsymbol{h}_i$ and its positive sample $\boldsymbol{h}_i^+$) as well as negative samples $(\boldsymbol{h}_{i1}^-, \cdots, \boldsymbol{h}_{iN}^-)$, which are then used to compute the $c$-th positive and negative temperatures. The computed temperatures and features of all positive and negative pairs are fed into the loss functions as in Table 2.

$$\Omega(\tau) = (d'/2)\log(\tau) + 1/\tau, \qquad (4)$$

which encourages temperature $\tau$ to move towards $\tau = 2/d'$. In Eq. (2.1), the asterisk '*' in variable $\tau_{in*}^{c-}$ indicates the index $n^* = \arg\max_{n=1,\cdots,N}\text{sim}(\boldsymbol{z}_i, \boldsymbol{z}_{in}^-)$. In Eq. (2.2), $\tau_i^{c+} = \sigma\big(\langle\phi(\boldsymbol{z}_i^c), \phi([\boldsymbol{h}_i^+]_{\text{sg}})\rangle\big)$ and $\tau_i^{\widetilde{c+}} = \sigma\big(\langle\phi(\boldsymbol{z}_i^{c+}), \phi([\boldsymbol{h}_i]_{\text{sg}})\rangle\big)$. In Eq. (2.3), temperatures are formed as $\tau_l^{c+} = \sigma\big(\langle\phi(\boldsymbol{z}_{l:}^c), \phi(\boldsymbol{z}_{l:}^{c+})\rangle\big)$

---

[1]This MLP layer is separate from the $C$ MLP projection heads.

and $\tau_{lm}^{c-} = \sigma\big(\langle\phi(\boldsymbol{z}_{l:}^c), \phi(\boldsymbol{z}_{m:}^{c+})\rangle\big)$ where $l \neq m$ and operator ':' simply indexes and concatenates variable as $\boldsymbol{z}_{l:} = [z_{l1}, \cdots, z_{lN}]^T$. In Eq. (2.4), $\boldsymbol{z}_{in}^{c\pm} = \boldsymbol{z}_{in}^{c-}$ and $\tau_{in}^{c\pm} = \tau_{in}^{c-}$ for $n = 1, \cdots, N$, and $\boldsymbol{z}_{in}^{c\pm} = \boldsymbol{z}_i^{c+}$ and $\tau_{in}^{c\pm} = \tau_i^{c+}$ if $n = N+1$. Finally, notice that for NT-Xent in Eq. (2.1), we have:

$$\frac{1}{\tau_{in^*}^{c-}} \max_{n=1,\cdots,N} \text{sim}(\boldsymbol{z}_i^c, \boldsymbol{z}_{in}^{c-}) - \Omega(\tau_{in^*}^{c-}) - (2\pi)^{d'/2} \approx \log \sum_{n=1}^N \frac{1}{(2\pi)^{d'/2}(\tau_{in}^{c-})^{d'/2}} \exp\left(\frac{1}{\tau_{in}^{c-}}\big(\text{sim}(\boldsymbol{z}_i^c, \boldsymbol{z}_{in}^{c-}) - 1\big)\right). \tag{5}$$

The same approximation (with $\boldsymbol{z}_{in}^{c\pm}$ in place of $\boldsymbol{z}_{in}^{c-}$) holds for InfoNCE in Eq. (2.4). Using maximum in Eq. (5), Eq. (2.1) and Eq. (2.4) is somewhat restrictive as the soft-maximum, depending on the temperature, will return the maximum similarity or interpolation over top few similarities close to maximum. Thus, soft-maximum will tackle a group of negative samples closest to the anchor. Indeed, we could use the above soft-maximum in place of maximum but then we have no easy way of recovering the temperature $\tau_{in^*}^{c-}$. Thus, in our experiments we observed that the best choice is to apply $\sum_{k=1}^\kappa \frac{1}{\kappa\tau_{in_k^*}^{c-}} \big[\text{TopMax}_\kappa \underset{n=1,\cdots,N}{\text{sim}}(\boldsymbol{z}_i^c, \boldsymbol{z}_{in}^{c-})\big]_k$ which is the average over top-$\kappa$ largest similarities. As the top-$\kappa$ maximum operation returns also corresponding indexes $n_1^*, \cdots, n_\kappa^*$, the temperature regularization can be easily computed as $\Omega\big(\{n_k^*\}_{k=1}^\kappa\big) = \log(\tau_{in_1^*}^{c-} \cdot \ldots \cdot \tau_{in_\kappa^*}^{c-}) + \sum_{k=1}^\kappa \frac{1}{\tau_{in_k^*}^{c-}}$.

## 4.2 DERIVING THE LOSS FUNCTION FROM MAXIMUM LIKELIHOOD ESTIMATION

This section details the derivation of our loss function based on the maximum likelihood estimation over head-wise posterior distributions of positive samples given observations. We show that our derivation is connected to an m-estimator (Huber et al., 1981) whose log-likelihood employs Normal distributions a.k.a. Welsch functions that are known to model the observation noise via the heteroscedastic aleatoric uncertainty (Matthies, 2007; Kendall & Gal, 2017; Hüllermeier & Waegeman, 2021). Our adaptive temperature captures such an uncertainty. Tuning constant $\tau$ was shown before to help learn good contrastive representations (Chen et al., 2020; He et al., 2020). Wang & Liu (2021) also demonstrated that temperature $\tau$ controls the strength of penalties on the hard negative samples and established its relationship with uniformity, illustrating that a well-chosen $\tau$ can strike a balance between the alignment and uniformity properties of contrastive loss. Kukleva et al. (2023) have shown that in place of constant temperature, a cosine schedule can improve learning–a seemingly minor modification with large impact on the learned embedding space.

For $\ell_2$ normalized vectors, the relationship between squared Euclidean distance $\|\cdot\|_2^2$ and cosine similarity measure is: $\|\boldsymbol{z}_i - \boldsymbol{z}_j\|_2^2 = 2 - \text{sim}(\boldsymbol{z}_i, \boldsymbol{z}_j)$. The Normal distribution $\mathcal{N}$ relies on the squared Euclidean distance. To derive our multi-head NT-Xent loss, consider the following maximum likelihood estimation *w.r.t.* parameters given as $\mathcal{P} = \big\{\boldsymbol{\theta}, \{\tau_i^{c+}\}_{c=1}^C, \{\{\tau_{in}^{c-}\}_{n=1}^N\}_{c=1}^C\big\}$ and $\beta = 1$:

$$\mathcal{P}^* = \arg\max_{\mathcal{P}} \prod_{c=1}^C \frac{\mathcal{N}\big(2 - 2\text{sim}(\boldsymbol{z}_i^c, \boldsymbol{z}_i^{c+}); \tau_i^{c+}\big)}{\sum_{n=1}^N \mathcal{N}\big(2 - 2\text{sim}(\boldsymbol{z}_i^c, \boldsymbol{z}_{in}^{c-}); \tau_{in}^{c-}\big)} \tag{6}$$

$$= \arg\min_{\mathcal{P}} \sum_{c=1}^C \left(-\log \mathcal{N}\big(2 - 2\text{sim}(\boldsymbol{z}_i^c, \boldsymbol{z}_i^{c+}); \tau_i^{c+}\big) + \log \sum_{n=1}^N \mathcal{N}\big(2 - 2\text{sim}(\boldsymbol{z}_i^c, \boldsymbol{z}_{in}^{c-}); \tau_{in}^{c-}\big)\right)$$

$$= \arg\min_{\mathcal{P}} \sum_{c=1}^C \left(-\frac{1}{\tau_i^{c+}}\text{sim}(\boldsymbol{z}_i^c, \boldsymbol{z}_i^{c+}) + \beta\Omega(\tau_i^{c+}) + \log \sum_{n=1}^N \frac{1}{(2\pi)^{d'/2}(\tau_{in}^{c-})^{d'/2}} \exp\left(\frac{1}{\tau_{in}^{c-}}\big(\text{sim}(\boldsymbol{z}_i^c, \boldsymbol{z}_{in}^{c-}) - 1\big)\right)\right). \tag{7}$$

In Eq. (7), we simply use expansion:

$$-\log\left(\frac{1}{(2\pi)^{d'/2}(\sigma^2)^{d'/2}} \exp\left(-\frac{2 - 2\mathbf{s}}{2\sigma^2}\right)\right) = d'/2\log(2\pi) + (d'/2)\log(\sigma^2) + 1/\sigma^2 - \mathbf{s}/\sigma^2, \tag{8}$$

where variance $\sigma^2 = \tau$. We drop the constant (no impact on optimization) and are left with $-\mathbf{s}/\tau$ and $\Omega(\tau) = (d'/2)\log(\tau) + 1/\tau$. We apply approximation in Eq. (5) to Eq. (7) (rightmost part) and readily obtain Eq. (2.1). To derive multi-head InfoNCE loss, we solve a slightly modified problem:

$$\mathcal{P}^* = \arg\max_{\mathcal{P}} \prod_{c=1}^C \frac{\mathcal{N}\big(2 - 2\text{sim}(\boldsymbol{z}_i^c, \boldsymbol{z}_i^{c+}); \tau_i^{c+}\big)}{\mathcal{N}\big(2 - 2\text{sim}(\boldsymbol{z}_i^c, \boldsymbol{z}_i^{c+}); \tau_i^{c+}\big) + \sum_{n=1}^N \mathcal{N}\big(2 - 2\text{sim}(\boldsymbol{z}_i^c, \boldsymbol{z}_{in}^{c-}); \tau_{in}^{c-}\big)} \tag{9}$$

$$= \arg\max_{\mathcal{P}} \prod_{c=1}^C \frac{\mathcal{N}\big(2 - 2\text{sim}(\boldsymbol{z}_i^c, \boldsymbol{z}_i^{c+}); \tau_i^{c+}\big)}{\sum_{n=1}^{N+1} \mathcal{N}\big(2 - 2\text{sim}(\boldsymbol{z}_i^c, \boldsymbol{z}_{in}^{c\pm}); \tau_{in}^{c\pm}\big)} = \arg\max_{\mathcal{P}} \prod_{c=1}^C p(\boldsymbol{z}_{ic}^{c+}|\boldsymbol{z}_i^c) = \arg\max_{\mathcal{P}} \prod_{c=1}^C \frac{p(\boldsymbol{z}_i^c|\boldsymbol{z}_{ic}^{c+})p(\boldsymbol{z}_{ic}^{c+})}{p(\boldsymbol{z}_i^c)},$$

where $p(\boldsymbol{z}_i^c) = \sum_{n=1}^{N+1} \mathcal{N}\big(2 - 2\mathrm{sim}(\boldsymbol{z}_i^c, \boldsymbol{z}_{in}^{c\pm}); \tau_{in}^{c\pm}\big)$, $p(\boldsymbol{z}_{ic}^{c+})$ is a constant, $e.g.$, 1, and $p(\boldsymbol{z}_i^c|\boldsymbol{z}_{ic}^{c+}) = \mathcal{N}\big(2 - 2\mathrm{sim}(\boldsymbol{z}_i^c, \boldsymbol{z}_i^{c+}); \tau_i^{c+}\big)$. Thus, the ratio of Gaussians in Eq. (9) can be interpreted as maximizing head-wise posterior distributions of positive samples given observations.

## 4.3 Discussion

**Why not use multiple backbones to improve feature learning?** This idea was explored in supervised learning (Tao, 2019). However, training multiple backbones imposes prohibitive computational costs in SSL with no guarantee on complementarity of such backbones.

**Connecting uncertainty to temperature.** Eq. 6 uses the variance $\tau$ of the distribution of pair-wise distances. Eq. 6 derives Eq. 7, where $\tau$ weighs the similarity, making it effectively the temperature. Because variance is usually treated as uncertainty (Zhang et al., 2021; Wang & Koniusz, 2022), we build natural correspondence between uncertainty and temperature.

**Connecting our loss function to existing contrastive learning methods.** Our loss function consists of three terms: (i) positive temperature-weighted similarities for positive pairs (ii) negative temperature-weighted similarities for negative pairs, and (iii) a regularization term for positive and negative temperatures. As identified in previous works (Wang & Isola, 2020; Wang & Liu, 2021), the alignment (closeness) of features from positive pairs and the uniformity of the induced distribution of the (normalized) features on the hypersphere are the two key properties in contrastive loss. Our loss function also optimizes these properties and further improves the contrastive learning performance through parameterized pair-wise temperature via re-weighting the positive and negative similarities. Our loss function is a more general form, and when we set the temperature to be a global constant, the constant regularization term no longer affect optimization, and thus the loss function reduces to the traditional contrastive loss.

**Physical meaning of the regularization term** $\Omega(\cdot)$ **in Eq. (3).** The regularization term consists of regularizing both the positive and negative temperatures. During optimisation, the $\log \tau$ term encourages lower positive temperatures and higher negative temperatures, whereas the reverse function term $\frac{1}{\tau}$ is in favour of higher positive temperatures and lower negative temperatures. Hence this regularization term balances the learning of positive and negative temperatures. As we jointly optimize network parameters and the temperature via MLE, this problem naturally becomes decomposed in the maximization of similarities for positive pairs (minimization for negative pairs) weighted by the temperature. However, if $\tau$ was to reach 0 for positive pairs, one would attain a trivial solution. $\Omega(\cdot)$ prevents that trivial solution. Intuitively, one can be very certain in similarity of sample pair but there is a price to pay for that certainty, imposed by $\Omega(\cdot)$ resulting from the Welsch function. This $\Omega(\cdot)$ expresses the prior belief or preference on temperature values.

**Adaptive temperature vs. attention learning.** The latter assigns varying weights to different components or parts of an object according to a specific design (Bahdanau et al., 2015; Chorowski et al., 2015; Caron et al., 2021). The learnable positive and negative temperatures reweigh the similarities by considering diverse image content resulting from multiple augmentations. This correction replaces the global temperature, allowing the backbone and multiple projection heads to focus on capturing different aspects of image content. Moreover, pair-wise weighted similarities on 'alignment' and 'uniformity' allow various similarity relations to contribute differently to contrastive learning, similar to an attention learning mechanism.

## 5 Experiments

We choose popular datasets that are widely used in evaluating the SSL models, including CIFAR-10, CIFAR-100, STL-10, Tiny-ImageNet and ImageNet. The dataset details are provided in Appendix A. Below, we describe the experimental setup and evaluation protocols.

### 5.1 Setup

We conduct experiments on the aforementioned datasets following practices outlined by Huang et al. (2022a). We consider five different types of transformations for data augmentations: random cropping, random Gaussian blur, color dropping ($i.e.$, randomly converting images to grayscale), color distortion, and random horizontal flipping. For pre-training, we employ ResNet18 (R18) (He et al.,

Table 3: Impact of AMCL when applied to popular and state-of-the-art SSL methods. On CIFAR-10, CIFAR-100, Tiny-ImageNet, and ImageNet, models are first pretrained for 1,000, 1,000, 800, and 100 epochs, respectively and then evaluated using linear probing. Backbones are highlighted.

| Datasets | | SimCLR | | MoCo | | SimSiam | | B.Twins | | CAN | | LGP | | Avg. gain |
|---|---|---|---|---|---|---|---|---|---|---|---|---|---|---|
| CIFAR-10 | baseline | 89.9 | R18 | 90.4 | R18 | 90.7 | R18 | 87.4 | R18 | - | | - | | ↑**2.43** |
| | ours | **92.2** | R18 | **92.9** | R18 | **93.0** | R18 | **90.0** | R18 | - | | - | | |
| CIFAR-100 | baseline | 57.6 | R18 | 64.4 | R18 | 63.6 | R18 | 58.2 | R18 | - | | - | | ↑**4.58** |
| | ours | **61.8** | R18 | **69.3** | R18 | **68.9** | R18 | **62.1** | R18 | - | | - | | |
| Tiny-ImageNet | baseline | 48.1 | R50 | 46.4 | R50 | 46.7 | R50 | 46.8 | R50 | 53.2 | ViT-B | 56.7 | ViT-B | ↑**1.65** |
| | ours | **50.0** | R50 | **47.8** | R50 | **49.0** | R50 | **48.3** | R50 | **54.9** | ViT-B | **57.8** | ViT-B | |
| ImageNet | baseline | 66.5 | R50 | 67.4 | R50 | 68.1 | R50 | 70.0 | R50 | 70.5 | ViT-B | 73.8 | ViT-B | ↑**1.67** |
| | ours | **68.1** | R50 | **69.3** | R50 | **69.6** | R50 | **72.7** | R50 | **71.6** | ViT-B | **75.0** | ViT-B | |

2016) for SimCLR (Chen et al., 2020), MoCo (He et al., 2020), SimSiam (Chen & He, 2021), and Barlow Twins (Zbontar et al., 2021) on CIFAR-10, CIFAR-100 and STL-10. For Tiny-ImageNet and ImageNet, we use ResNet50 (R50) variants, ViT-B and ViT-L. Other settings, such as the architecture of the projection head, remain consistent with the original algorithm configurations. We select $C$ projection heads in range from 2 to 6. We set the temperature bounds ($\eta$ and $\iota$ of sigmoid) in range $[1e-5, 2]$ on smaller datasets and $[1e-5, 5]$ for ImageNet and Tiny-ImageNet. For Barlow Twins, the regularization parameter $\lambda = 5e-3$. The temperature regularization parameter $\beta$ is varied from $1e-5$ to 10. Each model is trained with a batch size of 512 and 1000 epochs for small datasets, *e.g.*, STL-10; for large-scale datasets, *e.g.*, Tiny-ImageNet and ImageNet, we train for up to 800 epochs. To assess the quality of the encoder, we follow the KNN evaluation protocol (Wu et al., 2018) on small datasets. For large datasets, we use linear probing. For MIM-based methods such as CAN[2] and LGP[3], we select the standard ViT-B and ViT-L as the backbone encoder, with a token size $16 \times 16$. Other settings, including projection head architecture and hyperparameters, followed the original algorithm configurations. Both models are evaluated using a linear probe.

## 5.2 EVALUATION

**AMCL consistently improves popular and state-of-the-art SSL methods.** We apply AMCL to SimCLR, MoCo, SimSiam, Barlow Twins, CAN, and LGP. As shown in Table 3, on CIFAR-10, CIFAR-100, Tiny ImageNet and ImageNet datasets, average improvements across the baselines are 2.43%, 4.58%, 1.65%, 1.67%, respectively. In the case of MIM-based methods such as CAN and LGP, our multi-head approach improves them by 1.7% and 1.1%, respectively, on Tiny-ImageNet. The improvements over CAN and LGP on ImageNet are 1.1% and 1.2%, respectively.

**Effectiveness of AMCL under different backbones and training epochs.** We evaluate model capacity using the ImageNet dataset. We choose ResNet-50 with three different hidden layer widths

---

[2]https://github.com/bwconrad/can

[3]https://github.com/VITA-Group/layerGraftedPretraining_ICLR23

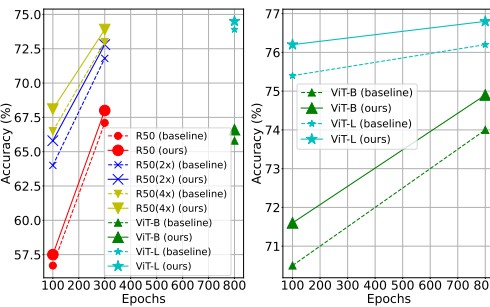

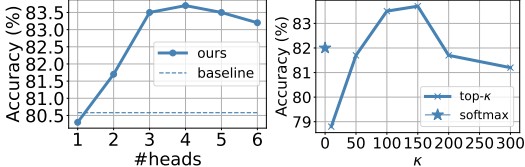

Figure 3: Impact of different backbones and training epochs on AMCL for (left) SimCLR and (right) CAN on the ImageNet dataset. All the reported accuracies use a linear probe.

Figure 4: Hyperparameter sensitivity analysis. (**left**:) number of projection heads. (**right**:) $\kappa$ in the selection of top-$\kappa$ similarities among negative pairs on STL-10. "Softmax" means using all negative pair similarities as in Eq. (7). We use ResNet18 as backbone with SimCLR. The dashed line is the baseline result with one projection head and global temperature .

Table 4: Comparing adaptive temperature with two state-of-the-art temperature methods on STL-10. Baseline (Base.) uses one projection head and global temperature, and the rest methods 3 heads. TaU (Zhang et al., 2021) views temperature as uncertainty, and TS (Kukleva et al., 2023) uses cosine schedule for temperature.

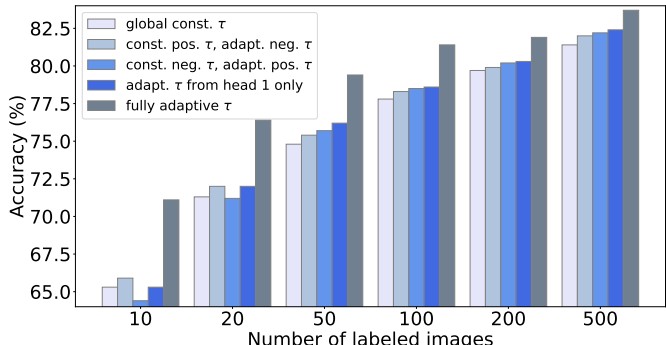

|     | Base. | ours | TaU | TS |
|-----|-------|------|-----|-----|
| 10  | 61.0  | **71.1** | 64.3 | 64.3 |
| 20  | 68.5  | **76.4** | 70.4 | 71.3 |
| 50  | 73.5  | **79.4** | 74.6 | 76.4 |
| 100 | 76.5  | **81.4** | 77.7 | 78.4 |
| 200 | 78.6  | **81.9** | 79.4 | 80.0 |
| 500 | 80.6  | **83.7** | 80.0 | 81.7 |

Figure 5: Comparing variants of adaptive temperature on STL-10. The pre-trained model is linear-probed with various numbers of labeled data. Number of heads is 3. "Const. pos. $\tau$, adapt. neg. $\tau$" uses constant temperature for positive pairs and adaptive for negative ones. Analogy goes for "Const. neg. $\tau$, adapt. pos. $\tau$". "Adapt. $\tau$ from head 1 only" copies adaptive temperatures from the first head to the other heads.

(width multipliers of $1\times$, $2\times$, and $4\times$), ViT-B, and ViT-L, which are widely used in SSL. In Fig. 3, under each epoch number, our AMCL yields consistent improvement to various backbones for SimCLR and CAN. Additionally, we observe that model capacity heavily depends on the choice of backbone. For ViT-L backbone encoder, SimCLR achieves the highest linear probe performance on ImageNet at $73.9\%$. When combined with AMCL, accuracy of SimCLR further increases by $0.6\%$.

**Impact of the number of projection heads** is presented in Fig. 4 (left) on the STL-10 dataset. ResNet18 is used as backbone, coupled with SimCLR. Adaptive temperature is always used. On STL-10, 3-5 projection heads are most effective, while other numbers also beat the baseline (except for $C = 1$). Considering the performance gain and the computational cost, we use 3 heads in our paper. Note that this is not a hyperparameter selection process but rather hyperparameter sensitivity evaluation. Instead, we choose hyperparameters using Hyperopt (Bergstra et al., 2015) on the validation set of each dataset.

**TopMax-$\kappa$ vs. Softmax in loss function.** In Fig. 4 (right), we compare using Top-$\kappa$ and Softmax when selecting negative pairs. Using Top-$\kappa$ is slightly better than Softmax when $\kappa = 100, 150$. In practice, $\kappa$ is chosen by Hyperopt (Bergstra et al., 2015) on the validation set of each dataset.

**Impact of number of labeled data for linear probing.** We train a logistic regression model on the STL-10 dataset with various numbers of labeled images: 10, 20, 50, 100, 200, and all 500 examples per class. We also use SimCLR for pre-training and linear-probe a standard ResNet-18 model (with random initialization) on the labeled training set of STL-10 (500 samples per class). Results are presented in Table 4 and Fig. 5. We have two observations. First, AMCL consistently improves SimCLR under different numbers of labeled data. Under 10, 50, and 200 labeled images, the improvement is 10.1%, 5.9%, and 3.3%, respectively. Second, compared with the fully supervised baseline with 73.3% accuracy, linear probing with 20 samples per class (1/25 of the whole labeled data) with our method is already superior (76.4%). This clearly demonstrates the advantages of self-supervised pretraining and our method.

**Comparing with temperature variants and state-of-the-art temperature schemes.** In Fig. 5, we compare the proposed adaptive temperature (fully adaptative $\tau$) with four variants, including making temperature of negative/positive pairs constant, copying adaptive temperature from one head to the others, and having global temperature. It is clear from the figure that our method is the best. In Table 4, we compare adaptive temperature with temperature as uncertainty (TaU) (Zhang et al., 2021) and temperature cosine schedule (TS) (Kukleva et al., 2023) under multiple heads, where adaptive temperature is also superior. In fact, the cosine temperature scheme is not adaptive to individual pairs, while 'temperature as uncertainty' is directly dependent on features not similarity. Being adaptive to individual pairs, their similarity and projection heads, our adaptive temperature is very well optimized under the derived loss function and thus exhibits very competitive performance.

| Augmentations | | | | | Accuracy | | | | Avg. |
|---|---|---|---|---|---|---|---|---|---|
| (a) | (b) | (c) | (d) | (e) | SC | MC | SS | BT | gain |
| ✓ | | | | | base. 26.3 | 39.9 | 26.4 | 33.7 | ↑**0.69** |
| | | | | | ours **27.0** | **40.3** | **27.0** | **34.7** | |
| ✓ | ✓ | | | | base. 28.0 | 40.2 | 26.6 | 34.2 | ↑**0.80** |
| | | | | | ours **29.0** | **40.9** | **27.3** | **35.1** | |
| ✓ | ✓ | ✓ | | | base. 44.4 | 57.3 | 51.5 | 49.8 | ↑**2.74** |
| | | | | | ours **46.2** | **61.0** | **55.4** | **51.3** | |
| ✓ | ✓ | ✓ | ✓ | | base. 55.4 | 62.7 | 60.7 | 55.0 | ↑**4.12** |
| | | | | | ours **58.5** | **66.3** | **67.0** | **58.4** | |
| ✓ | ✓ | ✓ | ✓ | ✓ | base. 57.6 | 64.4 | 63.6 | 58.2 | ↑**4.59** |
| | | | | | ours **61.8** | **69.3** | **68.9** | **62.1** | |

Table 5: Comparing AMCL with baselines under various numbers of augmentations. SC, MC, SS and BT denote SimCLR, MoCo, Sim-Siam and Barlow Twins, respectively. We report linear probing accuracy on CIFAR-100, where (a), (b), (c), (d), and (e) correspond to random cropping, random Gaussian blur, color dropping (*e.g.*, randomly converting images to grayscale), color distortion, and random horizontal flipping, respectively. ✓ denotes the corresponding augmentation is applied. Average improvement is shown in red.

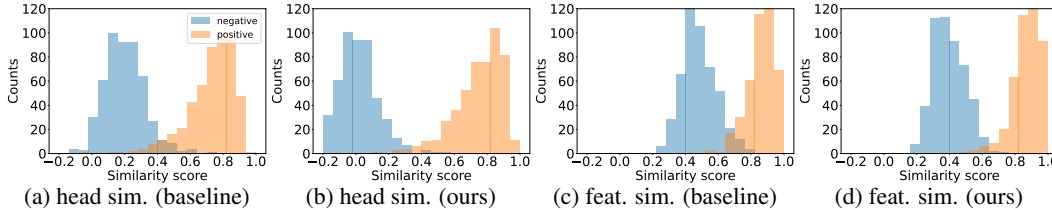

(a) head sim. (baseline)    (b) head sim. (ours)    (c) feat. sim. (baseline)    (d) feat. sim. (ours)

Figure 6: Distribution of similarity scores for positive and negative pairs. The baseline uses one projection head and global temperature, while our method has three projection heads and adaptive temperature. We use SimCLR for pre-training with the ResNet-18 backbone on STL-10. After pre-training, we choose 500 positive pairs and 500 negative pairs from the validation to compute the cosine similarity. In (a) and (b), similarity score (temperature scaled) is computed between the 128-dim features extracted from the projection head (s). In (c) and (d), cosine similarity score is computed between the 512-dim features extracted from the backbone.

**Impact of the number of data augmentation types.** In Table 5, under only 1-2 augmentations during SSL pretraining, the improvements in linear probing over the baselines are around 1%. When we further increase the number of augmentations, linear probing performance improves; importantly, AMCL becomes more and more useful: average improvement becomes as large as 4.59% when five types of data augmentation are used. This suggests the existence of multiple similarity relations when many data augmentations are applied, validating our motivation and method design.

**Visualization of pair similarity distributions.** In Fig. 6, we draw the similarity distributions of negative pairs and positive pairs, under the baseline (1 head + global temperature) and our method (multiple heads + adaptive temperature). When we use the average similarity across the output from the multiple heads, shown in Fig. 6(a) and (b), we can clearly observe better separability brought by our method. It indicates that our method allows for more effective similarity learning of the positive and negative pairs. On the other hand, if we compute the cosine similarity between features extracted right after the backbone, shown in Fig. 6(c) and (d), better separability can again be observed. It illustrates that better similarity learning further benefits representation learning, finally leading to improved linear probing proformance.

## 6 CONCLUSION

We are motivated by the complex pair similarity distributions under multiple augmentation types. We introduce adaptive multi-head contrastive learning (AMCL), which leverages multiple projection heads, each generating a distinct set of features, and a pair-wise adaptive temperature scheme. We derive our loss function and provide interesting insights such as the relationship between the variance of pair distance distribution and temperature, as well as physical meanings of regularization term. We show pair similarity distribution is better separated with our method. AMCL can be applied to and experimentally improve popular SSL methods with various backbones, numbers of labeled samples for linear probing, and augmentation types. AMCL is particularly useful under multiple augmentation types, consistent with our motivation.

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

# ADAPTIVE MULTI-HEAD CONTRASTIVE LEARNING —APPENDICES—

**Anonymous authors**

## A    DATASET DETAILS

We choose popular datasets that are widely used in evaluating the SSL models.

**CIFAR-10** (Krizhevsky, 2009) consists of 60,000 $32 \times 32$ colour images divided into 10 classes, each containing 6,000 images. The dataset is split into 50,000 training images and 10,000 test images.

**CIFAR-100** is similar to CIFAR-10 but comprises 100 classes, each with 600 images. There are 500 training images and 100 testing images per class. The 100 classes in CIFAR-100 (Krizhevsky, 2009) are grouped into 20 superclasses. Each image is labeled with both a 'fine' label (indicating its specific class) and a 'coarse' label (indicating its superclass).

**STL-10**  (Coates et al., 2011) is similarly to CIFAR-10 and includes images from 10 classes: airplane, bird, car, cat, deer, dog, horse, monkey, ship, truck. This dataset is relatively large and features a higher resolution ($96 \times 96$ pixels) compared to CIFAR10. It also provides a substantial set of $100,000$ unlabeled images that are similar to the training images but are sampled from a wider range of animals and vehicles. This makes the dataset ideal for showcasing the benefits of self-supervised learning.

**Tiny-ImageNet** (Le & Yang, 2015) contains 100,000 images of 200 classes (500 for each class) downsized to $64 \times 64$ colored images. Each class has 500 training images, 50 validation images, and 50 test images.

**ImageNet** (Deng et al., 2009) (a.k.a.**ImageNet-1K**) contains 14,197,122 annotated images according to the WordNet hierarchy. Since 2010 the dataset is used in the ImageNet Large Scale Visual Recognition Challenge (ILSVRC), a benchmark in image classification and object detection. The publicly released dataset contains a set of manually annotated training images.

