# OpenReview forum: "Adaptive Multi-head Contrastive Learning"
_ICLR.cc/2024/Conference — ICLR 2024 Conference Withdrawn Submission_

### Official Review · Reviewer_HgQy · 2023-10-21

**Soundness:** 2 fair
**Presentation:** 2 fair
**Contribution:** 2 fair
**Rating:** 3
**Confidence:** 4

**Summary:**

This paper proposes to enhance the contrastive learning framework with multiple projection heads and adaptive temperature.

**Strengths:**

The proposed multi-head strategy could be applied to various contrastive and non-contrastive learning frameworks. Experiments on four datasets demonstrate the effectiveness of the proposed method.

**Weaknesses:**

1. I do not quite get the novelty of this work. It seems that all this work has made is replicating the projection heads. The authors try to find a motivation from the similarity between positive and negative pairs to support such a modification. However, after reading the paper, I do not feel the observations in Fig. 1 correlate much with the proposed method.
2. As for the paper writing, probably due to the limited contributions in the method design, the authors spend a lot of space reviewing previous contrastive learning methods, which is not informative. The authors also provide the connection of the loss with the maximum likelihood estimation, which is also not novel and has few differences with existing literature.
3. Though the experiments show the proposed method enhances the performance, it is not surprising since it introduces more learnable parameters including the temperature, whose effectiveness has been proved in previous works.

**Questions:**

I hope the authors can address my concerns in the weakness section.

---

### Official Review · Reviewer_pd1e · 2023-10-28

**Soundness:** 2 fair
**Presentation:** 1 poor
**Contribution:** 2 fair
**Rating:** 3
**Confidence:** 4

**Summary:**

This paper proposes multiple projection heads for contrastive learning, to address the issue, where the similarity between augmented positive views is possibly lower than the one between negative views. It aims to improve the diversity of image representations by introducing more projection heads. In addition, it also presents an adaptive temperature to reweight the similarity between samples. The reported experimental results show the effectiveness of the proposed method.

**Strengths:**

+ This paper proposes a new problem, how to refine the similarity metric between positive- and negative- pairs. The plain contrastive learning methods address it by the fitting of global loss in the training dataset. This paper tries takeling it by multiple projection heads.

**Weaknesses:**

+ Empirically, multiple projection heads may not introduce more diversity of image representations, since a wider MLP projection head can achieve the similar target. A wider MLP head also provides larger capacity for image representations, whereas the results in SimCLR paper reports that wider MLP projection doesn't achieve more performance gain than the narrow one. So, as the wider MLP projection head, I don't think multiple projection heads can address the proposed problem.

+ The relationship between more diverities of image representation and inaccurate similarity problem in plain contrative methods (as clarified by athous) is not well clarified. The aim of projection head for contrastive objective is to capture the similarities but filter out the dissimilarities. Applying contrastive objective unavoidably removes some dissimilarities. More accurate learning targets may achieve better performance, instead of more projection heads.

+ The design of temperature in Eq.3 is not well clarified. As understood by me, this design is not reasonable. As Eq.3, a more similar pair is normalized by lager temperature. It causes that the temperature normalized similarity scores of all pairs, including positive- and negative- one, are the similar ones. How to discriminate positive pairs from negative ones.

+ The writing is somewhat confusing and can be furhter refined. The problems are listed in Question section.

+ The results of SimCLR and SimSiam on CIFAR10 are obvious worse than the ones reported in SimSiam paper (SimCLR: 89.9 vs 91.1; SimSiam: 90.7 vs 91.8). The experimental results are not so convicing for me.

**Questions:**

+ To verify the effectiveness of multiple projection heads, the authors should supplement comparative experiments between multiple projection heads and wider MLP projection head.

+ The architecture details of multiple projection heads is missing. They are very important experiment details for this paper.

+ The motivation of multiple projection heads and adaptive temperature should be clarified more clear. Why it works by analysis?

+ Eq.3 and Eq.4 are confusing. I also believe that Eq.4 is wrong.

+ For better convicing experimental reuslts, the authors should compare their results with the ones reported on the original paper.

+ The number of sampled pairs for Figure 6 should be more.

---

### Official Review · Reviewer_YrZy · 2023-10-31

**Soundness:** 3 good
**Presentation:** 3 good
**Contribution:** 3 good
**Rating:** 5
**Confidence:** 4

**Summary:**

This paper focuses on the false positive and false negative rooted in the contrastive learning paradigm derived from the ambiguous augmentations between positive pairs and semantic similarity between some with-class false negative pairs. To address such problems, this paper proposes using multiple projection heads and adaptive temperature to better capture the diverse image content and similarity relations under multiple augmentations. The authors applied the proposed adaptive multi-head contrastive learning (AMCL) approach to typical contrastive learning methods such as SimCLR, MoCo, and Barlow Twin, and the performance improvement validates the effectiveness of the proposed method.

**Strengths:**

This paper proposes a plug-and-played approach consisting of a multi-head projection strategy and adaptive temperature scaling regularization. The approach could be adopted into most contrastive learning methods, consistently improving the performance.

**Weaknesses:**

1. My major concern is the unclear mechanisms of the multi-head projection strategy and adaptive temperature scaling regularization. Although the effectiveness of the two modules has been empirically verified by the performance improvement against baseline, it is still not clear enough why they work. In other words, some discussion on the mechanisms of the two modules could be further clarified and some qualitative analysis or examples of how the multiple projection heads and adaptive temperature affect the learned representations and similarity scores could be supplemented.
2. This paper does not discuss the computational cost of using multiple projection heads and adaptive temperature, especially for large-scale datasets and models.
3. The paper introduces some crucial hyper-parameters in Equations 4 and 6. Such hyper-parameters are varied in different experiments. it's essential to include ablation studies to explore the sensitivity of the proposed method to variations in the hyper-parameters. A more comprehensive analysis would provide a deeper understanding of the method's robustness.

**Questions:**

Please see the weaknesses, thx